

# A new species of alien terrestrial planarian in Spain: *Caenoplana decolorata*

Eduardo Mateos[1], Hugh D. Jones[2], Marta Riutort[3] and Marta Álvarez-Presas[3,4]

[1] Departament de Biologia Evolutiva, Ecologia i Ciències Ambientals. Facultat de Biologia, Universitat de Barcelona, Barcelona, Spain
[2] Life Sciences Department, Natural History Museum, London, UK
[3] Departament de Genètica, Microbiologia i Estadística. Facultat de Biologia, Universitat de Barcelona, Barcelona, Spain
[4] School of Biological Sciences, University of Bristol, Bristol, UK

## ABSTRACT

Terrestrial planarians found in a plant nursery in Spain in 2012 are described as a new species, *Caenoplana decolorata*. Dorsally they are mahogany brown with a cream median line. Ventrally they are pastel turquoise fading to brown laterally. Molecular data indicate that they are a member of the genus *Caenoplana*, but that they differ from other *Caenoplana* species found in Europe. One mature specimen has been partially sectioned, and the musculature and copulatory apparatus is described, confirming the generic placement but distinguishing the species from other members of the genus. It is probable that the species originates from Australia.

## INTRODUCTION

*Álvarez-Presas et al. (2014)* recorded several terrestrial planarian species from Spain, some considered native to Europe, others introduced from other continents. Some species were identifiable on the basis of external features such as colour and shape, on anatomical characters and comparative molecular analysis. Molecular results suggested that further species were found but at the time they could not be certainly identified to species, though perhaps to genus. This paper describes specimens (Figs. 1A–1D) listed as '*Caenoplana* Ca2' by *Álvarez-Presas et al. (2014)*. Molecular data (Fig. 12 of *Álvarez-Presas et al., 2014*) indicate that these specimens are of the genus *Caenoplana Moseley, 1877*, but distinct from other *Caenoplana* species. One mature specimen has been partially sectioned, and the musculature and copulatory apparatus is described. It has the characters of the genus *Caenoplana Moseley, 1877*, as amended by *Ogren & Kawakatsu (1991)* and by *Winsor (1991)* but differs from other described species of that genus both in external characteristics and anatomy. Neither do the specimens resemble any species described only on external features such as shape and colour and currently placed in the genus *Australopacifica Ogren & Kawakatsu, 1991*, a collective genus containing species 'not classifiable into the present taxonomic genera because of insufficient morphological information; geographical distribution largely in Australasia and Indo-Pacific Islands.

Corresponding author
Eduardo Mateos, emateos@ub.edu

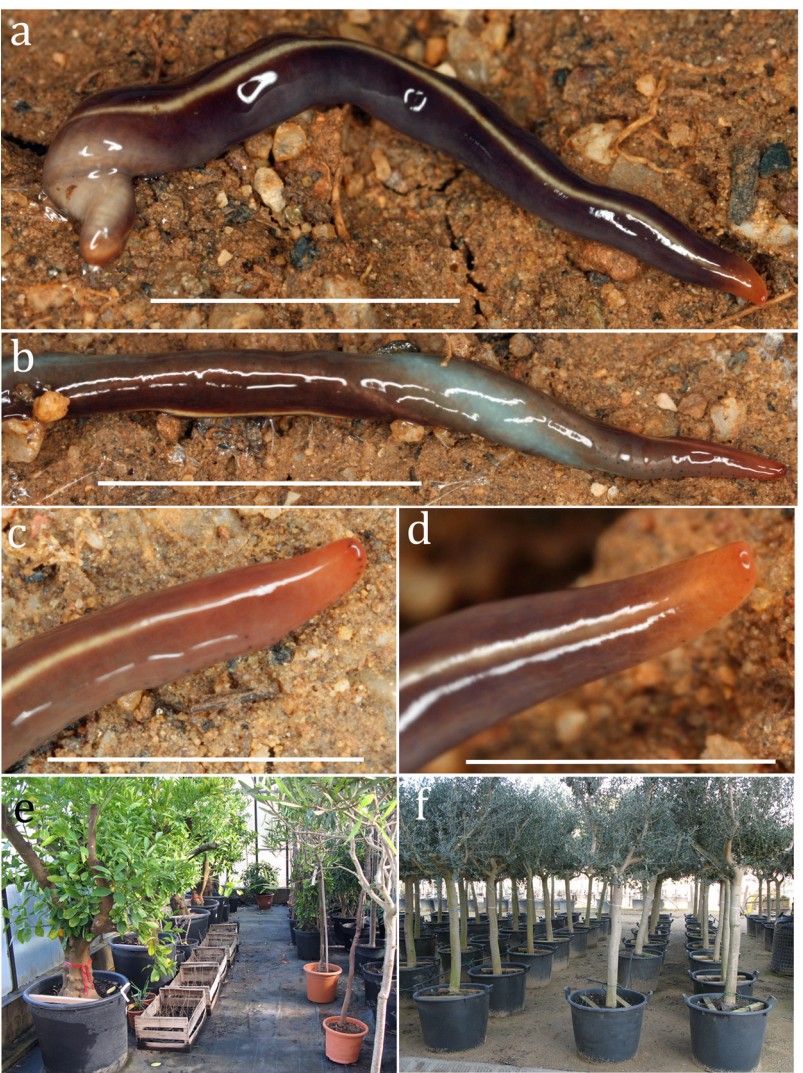

**Figure 1** ***Caenoplana decolorata* sp. nov.** (A–D) Photographs of live specimens, anterior to the right. (A) Dorsal view of specimen PT426 showing the 'mahogany brown' colour and 'cream' median line. Scale bar 10 mm. (B) A twisted specimen PT657-1 showing the 'pastel turquoise' ventral surface. Scale bar 10 mm. (C) Specimen PT657-1 and (D) specimen PT426, anterior end showing anterior 'copper brown' colour and the eyes (the two white lines in (D) are reflections from the lighting). Scale bars 4 mm. (E and F) Pots under which the specimens were found, in a greenhouse (E) and outdoors (F).

A collective group for *species inquirendae* and *nomina dubia*'. It is described as *Caenoplana decolorata* sp. nov.

## METHODS

### Sampling

Specimens were collected by E. Mateos from a plant nursery named "vivers casa Paraire" in Bordils municipality (Girona province, Spain, WGS84, position: 42.0348°N; 2.8986°E). All were collected by hand from beneath pots (Figs. 1E and 1F) that contained the

plants on 12 January 2012 (five specimens: PT426, PT427, PT428, PT430, PT431) and 22 October 2012 (four specimens: PT655, PT657-1, 2 and 3) (Table 1).

Specimens from 12 January 2012 and specimens PT655 and PT657-3 were preserved in absolute ethanol for further molecular analyses. Specimens PT657-1 and 2 were killed with boiling water, fixed with 10% formalin and preserved in 70% ethanol. Specimens PT426 and PT657-1 were photographed alive (Fig. 1).

## Molecular methods

All the sequences used in the present work were obtained in previous studies with the exception of some *Cytochrome Oxidase I* (herein Cox1) sequences that were obtained from individuals collected at the Real Jardín Botánico de Córdoba (Spain) by Mónica López (Table 1). In all cases, a small section of the anterior end of specimens preserved in absolute ethanol was used for DNA extraction. The new sequences were obtained following the same protocol as in *Álvarez-Presas et al. (2014)*.

A nucleotide alignment was obtained for the Cox1 sequences based on the AA translation as a guide using BioEdit software (*Hall, 1999*) and the echinoderm mitochondrial genetic code (9). A Maximum Likelihood (ML) phylogeny was inferred using the IQtree software (*Nguyen et al., 2015*) with the MFP+MERGE implementation and 10,000 replicates for ultrafast bootstrap search (-bb option). Then two single locus molecular species delimitation methods were applied in order to check the validity of the new species presented here and the ones already described and included in the phylogeny. Automatic Barcode Gap Discovery (ABGD) (*Puillandre et al., 2012*) was the first method performed, implemented in the webpage: https://bioinfo.mnhn.fr/abi/public/abgd/abgdweb.html. The default parameters were used, selecting initial partitions as they are supposed to be more stable and generally give as a result a closer number of groups described by taxonomists than recursive partitions. The second method applied was the multi-rate Poisson Tree Process (mPTP) analysis (*Kapli et al., 2017*). The newick tree obtained in the ML phylogenetic inference was used as input in the website http://mptp.h-its.org/#/tree.

## Anatomical methods

Specimens PT657-1 and 2 were sent to HDJ and are deposited in the Natural History Museum, London, accession numbers NHMUK.2014.5.13.12-13. The larger specimen had a visible gonopore, was assumed to be mature and selected for partial sectioning. It was divided into four portions: anterior portion about 2 cm long not sectioned, in alcohol; pre-pharyngeal section, TS, five slides, two at 15 μm, three at 10 μm; posterior portion including pharynx and copulatory apparatus, LS, 16 slides (pharynx separated, HLS) at 15 μm. Sectioned portions were dehydrated in ethanol and embedded in paraffin wax. Slides were stained in Harris's haematoxylin and eosin and mounted in Canada balsam. The second specimen, about 3.4 cm long, had no visible gonopore and remains in alcohol.

Colours are expressed as RAL colours (www.ralcolor.com).

The electronic version of this article in Portable Document Format (PDF) will represent a published work according to the International Commission on Zoological Nomenclature

**Table 1 List of samples used in the molecular analysis with GenBank accession numbers.**

| Species/morphotype | Locality | GenBank Code Cox1 |
|---|---|---|
| Family Geoplanidae | | |
| Subfamily Rhynchodeminae | | |
| Tribe Caenoplanini | | |
| *Artioposthia* sp. | Australia | MN990642 |
| *Arthurdendyus testaceus* | – | MN990643 |
| *Caenoplana coerulea* | New Zealand | DQ665961 |
| | Menorca (Spain) | JQ514564 |
| | Liverpool, UK | DQ666030 |
| | El Prat de Llobregat (Barcelona, Spain) | KJ659617 |
| | Vall de'n Bas (Girona, Spain) | KJ659618 |
| | | KJ659619 |
| | | KJ659620 |
| | | KJ659622 |
| | | KJ659623 |
| | | KJ659624 |
| | | KJ659626 |
| | Badalona (Barcelona, Spain) | KJ659633 |
| | | KJ659634 |
| | Adelaide (Australia) | KJ659642 |
| | – | KJ659645 |
| | Granollers (Barcelona, Spain) | KJ659647 |
| PT1304 | Real Jardín Botánico de Córdoba (Córdoba, Spain) | MT727076* |
| PT1305 | | MT727077* |
| PT1307 | | MT727078* |
| PT1310 | | MT727079* |
| *Caenoplana* sp. 1 | – | DQ666031 |
| *Caenoplana* sp. 2 | Tallaganda (Australia) | DQ227621 |
| | | DQ227625 |
| | | DQ227627 |
| | | DQ227634 |
| *Caenoplana* sp. 3 | Victoria (Australia) | DQ465372 |
| *Caenoplana* sp. 4 | – | DQ666032 |
| *Caenoplana variegata* | Bordils (Girona, Spain) | KJ659648 |
| | Southampton, UK | MN990646 |
| | Cardiff, UK | MN990647 |
| | | MN990648 |
| *Caenoplana decolorata* sp. nov. | Bordils (Girona, Spain) | KJ659628 |
| | | KJ659629 |
| | | KJ659630 |
| | | KJ659631 |
| | | KJ659632 |
| Table 1 (continued) | | |
|---|---|---|
| **Species/morphotype** | **Locality** | **GenBank Code** **Cox1** |
| | | MN990644 |
| | | KJ659649 |
| OUTGROUP: tribe Rhynchodemini | | |
| *Dolichoplana sp.* | – | DQ666037 |
| *D. striata* | Igreginha (Brazil) | KC608226 |
| *Rhynchodemus sylvaticus* | Canyamars (Barcelona, Spain) | FJ969946 |

**Note:**
\* Sequences obtained in this study.

(ICZN), and hence the new names contained in the electronic version are effectively published under that Code from the electronic edition alone. This published work and the nomenclatural acts it contains have been registered in ZooBank, the online registration system for the ICZN. The ZooBank Life Science Identifiers (LSIDs) can be resolved and the associated information viewed through any standard web browser by appending the LSID to the prefix http://zoobank.org/. The LSID for this publication is: urn:lsid:zoobank.org:pub:B2636DF8-4372-405C-8A8C-4FBEC7396276. The LSID for the new species described is: *Caenoplana decolorata* sp. nov. urn:lsid:zoobank.org:act: C0CEE92F-A51E-4EDD-B18B-E7F021338667. The online version of this work is archived and available from the following digital repositories: PeerJ, PubMed Central and CLOCKSS.

## RESULTS

### Molecular results

The final dataset comprises 43 Cox1 sequences (including three outgroups, Table 1), with a final length of 822 bp. The resulting ML tree (Fig. 2) shows monophyletic groups comprising seven putative *Caenoplana* species. Although the bootstrap values (bb) are not high enough to give support to the relationships between these clades, the monophyly of the new species described here, *C. decolorata*, harbor maximum support. The results of the molecular species delimitation analyses (both mPTP and ABGD) match the same clades present in the phylogeny (Fig. 2) giving rise to seven putative *Caenoplana* species. Among them, we find the subject of this study, *Caenoplana decolorata*.

### Taxonomic section

Order Tricladida *Lang, 1884*
   Suborder Continenticola *Carranza et al., 1998*
      Family Geoplanidae *Stimpson, 1857*
         Subfamily Rhynchodeminae *von Graff, 1896*
            Tribe Caenoplaninae *Ogren & Kawakatsu, 1991*
               Genus *Caenoplana Moseley, 1877*

Tree scale: 0.01 ⊢⊣

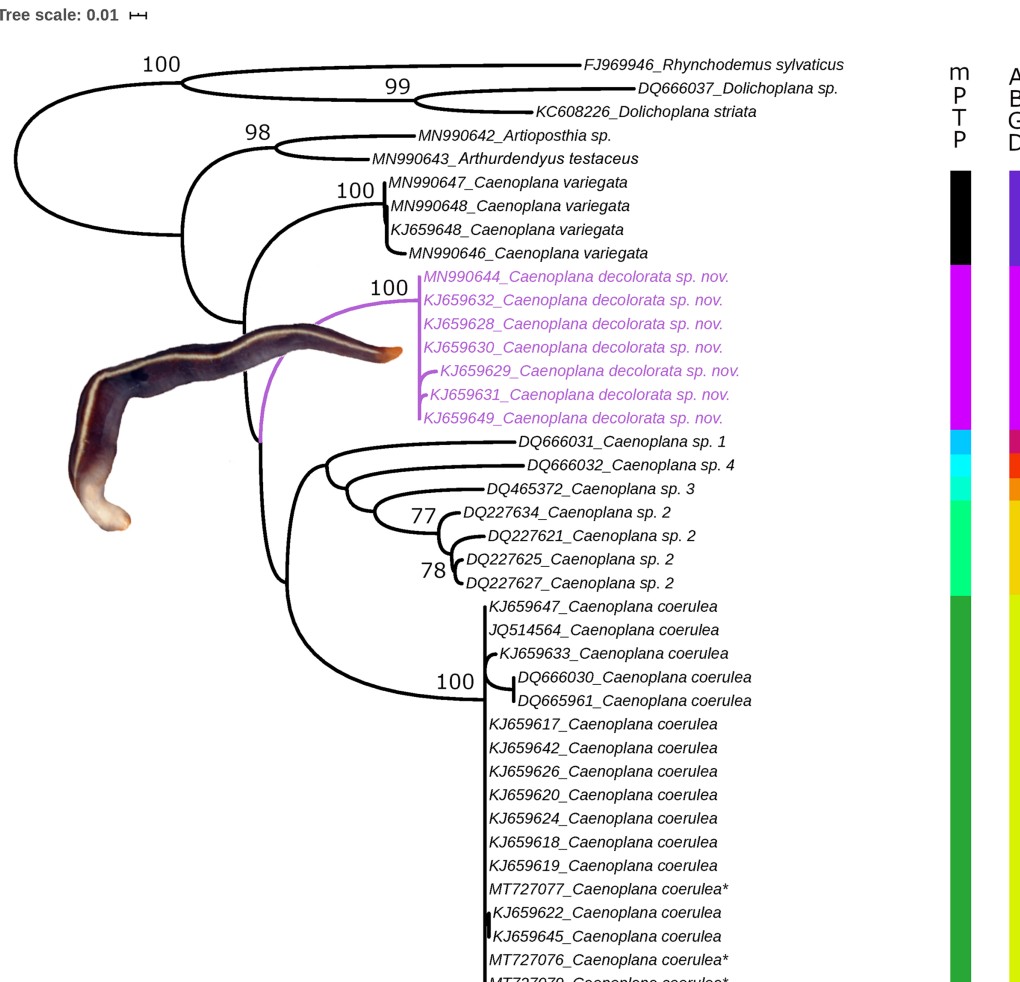

**Figure 2 Maximum Likelihood (ML) phylogeny inferred with Cox1 sequences.** Values at nodes correspond to ultrafast bootstrap replicates (bb) obtained with IQtree software. Vertical bars to the right of the phylogeny correspond to the molecular species delimitation methods results (mPTP, left bar and ABGD, right bar). Scale bar represents number of substitutions per site. Photograph of specimen PT426 in dorsal view (anterior to the right).

*Caenoplana decolorata* new species.

*Caenoplana* Ca2 *Álvarez-Presas et al., 2014*.

Etymology: "decolorata" indicating that live specimens resemble *C. coerulea* but are comparatively pale and discolored.

## NHMUK.2014.5.13.12-13

E. Mateos collection code PT657-1 and PT657-2. Locality: Bordils (Girona, Spain), position N42.0348049 E2.8986153, date 22 October 2012.

Preserved dimensions: holotype (PT657-1): length 46 mm; width 2 mm; height 1 mm; anterior to mouth 28 mm (61% of body length); anterior to gonopore 39 mm (85% of body

length); paratype (PT657-2): length 34 mm; width 2.1 mm; anterior to mouth 17 mm (50%); apparently immature.

All other specimens (with a small section of the anterior end removed) are deposited in the collection of M. Riutort at the University of Barcelona.

## External characters

Live specimens are 'mahogany brown' (RAL 8106) with a narrow 'cream' (RAL 9001) mid-line dorsally, merging to 'beige brown' (RAL 8024) laterally. The anterior end is 'copper brown' (RAL 8004). The ventral mid-line is 'pastel turquoise' (RAL 6034) merging into the lateral 'beige brown.'

Eyes in a sparse uniserial row round the anterior end, biserial for a short distance behind the anterior end and sparse staggered uniserial to the posterior end. Sole nearly the whole of the ventral surface.

## Anatomy

Transverse sections (Fig. 3A) are about 1.3 mm high and 2 mm wide. The ciliated creeping sole is about 80% of the width. The cilia are about 5 μm long. The ventral epidermis is a monolayer about 30 μm thick and has few rhabdites. Ventral sub-epidermal muscle consists of a layer of circular muscle fibres about 10 μm thick and longitudinal muscle in bundles about 30 μm thick. Dorsal to the longitudinal muscle bundles is a ventral nerve plexus. There is a distinct, compact layer of parenchymal longitudinal muscle ventrally, 40–50 μm thick, 150 μm in from ventral surface. Ventral nerve cords are about 750 μm centre to centre, about 120 μm in diameter, with transverse commissures. Laterally and dorsally the parenchymal longitudinal muscle is less compact and 10–20 μm thick. Dorsal epidermis is 45 μm thick, non-ciliated and has numerous rhabdites. Dorsal and lateral sub-epidermal circular muscle is about 10 μm thick, and longitudinal muscle in bundles about 35 μm thick. Rhabdites are numerous dorsally and laterally ental to the sub-epidermal muscle, but in the mid-dorsal region, the rhabdites layer is slightly deeper (Figs. 3A and 3B), presumed to be coincident with the pale midline visible in the living animal.

The retracted cylindrical pharynx occupies the whole length of the pharyngeal pouch and is about 2.5 mm long, 0.9 mm in diameter. The pouch is 5.4% of body length. The pharyngeal aperture is about half way along the pharyngeal pouch. Pharyngeal musculature consists of an outer layer of circular muscle about 10 μm thick, a layer of mixed longitudinal and radial muscle about 360 μm thick and an inner layer of circular muscle about 30 μm thick.

The anterior portion containing the ovaries has not been sectioned. Ovovitelline ducts are about 500 μm apart on the inner dorsal surface of the ventral nerve cords (Figs. 3A and 3D). Vitellaria are not distinguishable with certainty. Their outer and inner diameters are about 25 μm and 7 μm respectively. They run to about 800 μm behind the gonopore, turn dorsally and open into the common female duct about 800 μm long which extends forwards with little differentiation to open into the common antrum above the
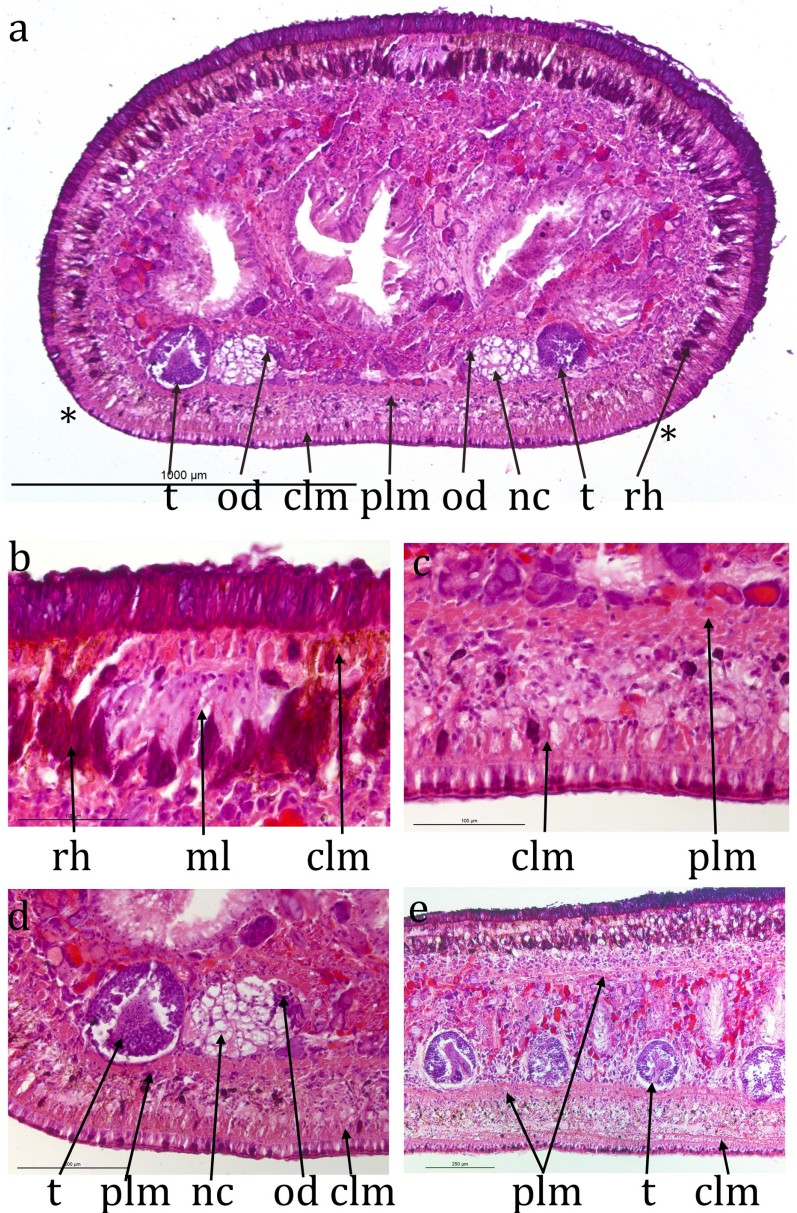

**Figure 3** *Caenoplana decolorata* specimen **PT657-1 (NHMUK2014.5.13.12).** (A) Entire transverse section (**indicate the width of the ventral creeping sole; scale line 1 mm). (B) Enlarged mid-dorsal (scale line 100 μm). (C) Enlarged mid-ventral (scale line 100 μm). (D) The testis, ventral nerve cord and ovovitelline duct on one side (scale line 200 μm). (E) Longitudinal section showing several testes (scale line 250 μm).

gonopore (Figs. 4A, 4C and 4E). There is little or no shell gland tissue surrounding the common female duct.

Testes are numerous, ventral, ovate, about 200 μm wide and 300 μm high (Figs. 4A, 4D and 4E) and run almost to the copulatory apparatus. The sperm ducts cannot be distinguished with certainty in transverse sections. They enter the anterior end of the muscular bulb of the eversible penis, widen slightly and contain small amounts of stored

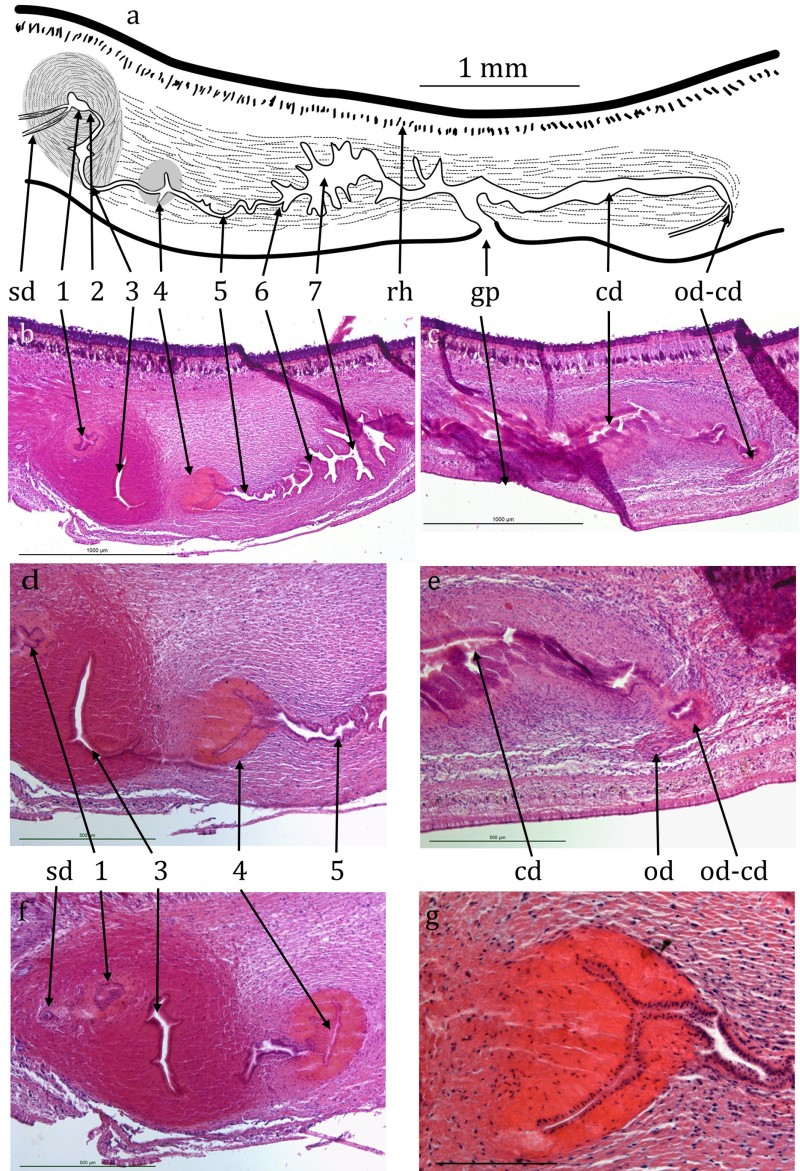

**Figure 4** *Caenoplana decolorata* specimen PT657-1 (NHMUK2014.5.13.12). (A) Reconstruction diagram and (D–G) longitudinal sections of the copulatory apparatus (anterior to the left); (A), (B) and (C) are to the same scale. Micrographs: (B and C) Mid-sections through the male and female portions respectively (both folded sections) (scale lines 1,000 μm). (D and F) Further sections through the proximal portion of the male ducts (scale lines 500 μm). (E) Section showing the approach of an ovovitelline duct to the common female duct (scale line 500 μm). (G) Enlargement of region 4 of the male duct (scale line 200 μm). The nuclei (cyanophil) are mostly adjacent to the lumen.

sperm (Fig. 4B). They separately enter the anterior end of the ejaculatory duct which is complex, long and sinuous, about 1.5 mm from its anterior end to the gonopore (Figs. 4A, 4B, 4D and 4F). It has several regions, for ease of reference they are here arbitrarily numbered 1–7 from anterior to posterior as follows. (1) A small chamber (seminal vesicle) which extends transversely through $10 \times 15$ μm sections, thus about 150 μm wide, the two

sperm ducts entering on either lateral extremity. (2) A narrow duct extending posteriorly and turning ventrally and opening into, (3) a sinus-like duct wide laterally, 23 × 15 µm thus 345 µm wide, but only 35 µm in the antero-posterior direction. This duct initially turns ventrally then narrows and curves posteriorly to be almost U-shaped (second arm shorter). The ejaculatory duct continues into, (4) a narrow sinus-like lumen surrounded by strongly eosinophilic cells forming a structure roughly spherical in outline about 400 µm in diameter. The cells of this region appear to be elongate with nuclei mostly adjacent to the lumen (Fig. 4G). This in turn opens into, (5) a portion about 400 µm long with sinuous margins, which in turn opens via, (6) a small papilla-like opening into, (7) a longer and wider duct about 600 µm long with sinuous walls which can be considered to be the male antrum. This in turn opens to the common antrum above the gonopore.

## DISCUSSION

The previous molecular results (Álvarez-Presas et al., 2014) analyzing only *Caenoplana* sequences (and an outgroup) indicated that *C. decolorata* specimens are closely related to *Caenoplana variegata* (Fletcher & Hamilton, 1888) (named as *C. bicolor* (von Graff, 1899) in that work, see Jones et al., 2020) although without support. In the present work, the tree shows a closer relationship between *C. decolorata* and *C. coerulea*, while *C. variegata* is sister to the clade formed by these two species (plus some putative unknown species), which will be an expected result having into account the more similar external coloration pattern of the first two species. However, the bb values do not support the relationships among species in the present work neither and make impossible to validate this hypothesis.

The sectioned specimen has multiple eyes, ventral testes, a layer of parenchymal longitudinal muscle, stronger ventrally, a long and fairly elaborate copulatory apparatus, the ejaculatory duct particularly so, and other anatomical characters of the genus *Caenoplana Moseley, 1877* as amended by Ogren & Kawakatsu (1991) and by *Winsor (1991)*. Thus we are confident of the generic placement.

However, comparison with other *Caenoplana* species is problematic. Ogren & Kawakatsu (1991) list 11 species of *Caenoplana* each with an anatomical description. Winsor (1991) lists 19 species, seven 'provisionally placed,' within *Caenoplana*. None of those has a similar external coloration to the present specimens, and the ejaculatory duct of the present specimens is distinctly different to that of any of those 11. They also differ from *C. variegata* (Fletcher & Hamilton, 1888) (synonymous with *C. bicolor* (von Graff, 1899), see Jones et al. (2020)).

Winsor (1997) lists a further six numbered, unnamed, *Caenoplana* species in addition to two named species, *C. coerulea coerulea* (Moseley, 1877) and *C. bicolor* (von Graff, 1899). Winsor (1998) states that 22 *Caenoplana* species were present in Australia, with no other details. Presumably this total includes the six numbered, unnamed, species above. Álvarez-Presas et al. (2014) list two further unnamed *Caenoplana* species, one the subject of this paper. Whether either of these is similar to any of *Winsor's (1997)* unnamed species is unknown.

In comparing this species to other *Caenoplana* species or to species placed in the collective genus *Australopacifica*, particular attention should be made to those with a broadly similar pigment distribution, that is those with, dorsally, a narrow mid-dorsal pale line on an otherwise uniform dark colour (any dark colour) and ventrally with a more or less uniform, but different, colour. The only two species with such a distribution are *C. coerulea* Moseley, 1877 and *C. purpurea* (Dendy, 1895).

*Caenoplana coerulea* Moseley, 1877, originally found in New South Wales, Australia, was described as follows: 'entire body of a dark Prussian blue colour, somewhat lighter on the under surface … with a narrow, mesial, dorsal, longitudinal stripe of white;' 5 cm long. Hyman (1943, 1954) and Ogren (1989) have described the anatomy of similar specimens found in the USA. This species has distinctly different coloration from the present specimens and the ejaculatory duct has a different structure (Ogren, 1989). It has subsequently been found in New Zealand (Dendy, 1895), several European countries (Álvarez-Presas et al., 2014) and North and South America (Ogren, 1989; Luis-Negrete, Brusa & Winsor, 2011).

*Geoplana purpurea* Dendy, 1895, originally from South Island, New Zealand, was described as follows: 'dorsal surface rather dark reddish-purple … a very narrow median band of nearly white,' 'ventral surface paler purple, under a lens appearing very finely mottled in two shades;' 3.5 cm long. Dendy (1895) comments: 'it is perhaps doubtful whether this species ought to be separated from the Australian *G. coerulea*, from which it differs only in colour.' But in the same paper Dendy also records *C. coerulea*. *Geoplana purpurea* was placed by Ogren & Kawakatsu (1991) in the collective genus *Australopacifica*, with the note that 'this probably belongs to *Caenoplana* on basis of external similarities to *Caenoplana coerulea*.' Winsor (1991) 'provisionally placed' it within *Caenoplana*. There has been no anatomical description of specimens under that species name. However, the coloration is different to the specimens from Spain and it seems unlikely that the latter are of the same species.

None of the other species listed by Ogren & Kawakatsu (1991) under *Australopacifica* has a colouration similar to the present species.

Thus the specimens do not match the description of any species previously described and are described as a new species, *Caenoplana decolorata*.

One possible confusing factor is that the colour of some species has been shown to vary over time and between individuals due to feeding (Jones et al., 2020; McDonald & Jones, 2007). Only prolonged observations on live animals before and after feeding could clarify if that might be the case with this species. Such observations would also indicate its preferred prey.

The ejaculatory duct of the new species is distinctive. The structure here numbered 4 is unlike anything present in any other described species of *Caenoplana* or for that matter any other terrestrial planarian. The function of this structure is not obvious; it does not appear to be either glandular or muscular.

Since at least one of the specimens was mature, it is presumed that this species reproduces by sexual reproduction, though it is entirely possible that it may also reproduce by partial fission, as in *C. variegata* (see Jones et al., 2020) and several other land planarian species.

This species almost certainly originates from Australia since most *Caenoplana* species are from there. It is presumed that it has been inadvertently transported to Spain in the course of the trade in horticultural products.

## FIGURE ABBREVIATIONS

| | |
|---|---|
| **1–7** | Arbitrary regions of the ejaculatory duct (see text) |
| **cd** | Common female duct |
| **gp** | Gonopore |
| **clm** | Cutaneous longitudinal muscle |
| **ml** | Median dorsal line |
| **nc** | Nerve cord |
| **od** | Ovovitelline duct |
| **od–cd** | Opening of ovovitelline ducts to common female duct |
| **plm** | Parenchymal muscle |
| **rh** | Rhabdites |
| **sd** | Sperm duct |
| **t** | Testis |

## ACKNOWLEDGEMENTS

We thank Mónica López, from the Real Jardín Botánico de Córdoba (Spain), for collecting and supplying some flatworm specimens from Córdoba. HDJ would like to thank The School of Biological Sciences, University of Manchester and Peter Walker of the histology laboratory, for access to facilities.

### Funding

This research was supported by the Ministerio de Ciencia, Innovación y Universidades, Spain (project 2018-PGC2018-093924-B-100). The funders had no role in study design, data collection and analysis, decision to publish, or preparation of the manuscript.

### Grant Disclosures

The following grant information was disclosed by the authors:
Ministerio de Ciencia, Innovación y Universidades, Spain: 2018-PGC2018-093924-B-100.

### Competing Interests

Marta Riutort is an Academic Editor for PeerJ.

### Author Contributions

- Eduardo Mateos conceived and designed the experiments, performed the experiments, analyzed the data, prepared figures and/or tables, authored or reviewed drafts of the paper, and approved the final draft.

- Hugh D. Jones conceived and designed the experiments, performed the experiments, analyzed the data, prepared figures and/or tables, authored or reviewed drafts of the paper, and approved the final draft.
- Marta Riutort conceived and designed the experiments, performed the experiments, analyzed the data, prepared figures and/or tables, authored or reviewed drafts of the paper, and approved the final draft.
- Marta Álvarez-Presas conceived and designed the experiments, performed the experiments, analyzed the data, prepared figures and/or tables, authored or reviewed drafts of the paper, and approved the final draft.

## Data Availability

Cox1 data is available at GenBank: MN990642, MN990643, DQ665961, JQ514564, DQ666030, KJ659617, KJ659618, KJ659619, KJ659620, KJ659622, KJ659623, KJ659624, KJ659626, KJ659633, KJ659634, KJ659642, KJ659645, KJ659647, DQ666031, DQ227621, DQ227625, DQ227627, DQ227634, DQ465372, DQ666032, KJ659648, MN990646, MN990647, MN990648, KJ659628, KJ659629, KJ659630, KJ659631, KJ659632, MN990644, KJ659649, DQ666037, KC608226, FJ969946.

## New Species Registration

The following information was supplied regarding the registration of a newly described species:

Publication LSID: urn:lsid:zoobank.org:pub:B2636DF8-4372-405C-8A8C-4FBEC7396276.

Caenoplana decolorata sp. nov.: urn:lsid:zoobank.org:act:C0CEE92F-A51E-4EDD-B18B-E7F021338667.

## Supplemental Information

Supplemental information for this article can be found online at http://dx.doi.org/10.7717/peerj.10013#supplemental-information.

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
