# Peer review of "A new species of alien terrestrial planarian in Spain: Caenoplana decolorata"

_PeerJ, doi:10.7717/peerj.10013_

## Round 0.1 · original submission · Minor Revisions

The description of this species as a nominal species will be important for future conservation purposes - administrations require binomial Latin names for lists of invasive species.

Both reviewers have given suggestions to improve the manuscript, but it seems clear that it needs only minor revision.

Additional comments. Note a minor mistake in the tree: "Arthurdendyus" (misspelled). The format of references seems odd to me, with authors in italics in the text, please check instructions of the journal.

·

Basic reporting

Clear professional English is used throughout the paper, and the introduction provides the necessary context for this work. The cited literature is relevant to this study, the structure conforms to PeerJ standards, including the data deposition statement, it complies with the new species policies, and meets the ICZN standard. The figures are of a high standard and are correctly labelled and described.

Experimental design

No comment

Validity of the findings

Both morphological and molecular data support the placement of the species in the genus Caenoplana and evidence is provided that the specimens belong to a new species C. decolorata. Although a specific morphological diagnosis is not provided, a differential diagnosis is provided in the discussion.

Additional comments

It has been stated* that “It is clear that the means to resolving many taxonomic problems lies with species descriptions and generic diagnoses with comparable treatment of comparable characters.” The recognition of so-called ”cryptic species” revealed by molecular studies, and later confirmed by detailed comparative morphological investigations may require the re-evaluation of characters not previously considered of taxonomic significance; this underlines the need for complete descriptions.

With this comment in mind it is unfortunate that the morphological account of C. decolorata is incomplete: the anterior portion of the Holotype was not sectioned but retained in alcohol, precluding the present description of, for example, anteriad sensory and reproductive structures. Furthermore, the morphological description provided in this paper seems compromised in part by the thickness of the sections (10 µm -15 µm thick); such thick sections appear to have limited the description of cytological details of the histology of the copulatory organs and frustrated determining the detailed histology of the ejaculatory duct, and in particular the nature of the enigmatic structure near position 4.

Minor corrections and suggestions are provided under Track Changes in the attached document that should be rectified before acceptance of this paper. It is also suggested that to facilitate comparisons of comparable characters across genera and species, that where indicated, the morphological description of the new species is harmonized with terminology and metrics generally currently provided elsewhere in the morphological descriptions of species od terrestrial flatworms.


*Winsor, Johns and Yeates (1998) Introduction, and ecological and systematic background, to the Terricola (Tricladida). Pedobiologia 42: page 400.

·

Basic reporting

No comment.

Experimental design

No comment.

Validity of the findings

No comment.

Additional comments

The manuscript deals with the description of a new land planarian of the genus Caenoplana that has been introduced into Europe, which is supported by both morphological and molecular data. This is an important work considering the still poor knowledge about the diversity of land planarians on a global scale. The manuscript is well-written, with proper English, follows the general structure of works on the taxonomy of land planarians and has adequate methods.

My main questions and suggestions are related to the description of the copulatory apparatus of the new species.

The authors describe the sperm ducts as entering the ejaculatory duct directly and consider the ejaculatory duct as having seven distinct regions. In most land planarians, the proximal part of the canal or cavity that the sperm ducts enter receives large quantities of secretion and is called seminal vesicle or prostatic vesicle. The regions 1-3 seem to be very similar to what is usually considered as the prostatic or seminal vesicle, so I am a bit intrigued about why the authors did not interpret this region as such considering that, according to Winsor’s thesis (2003) and the Ogren’s (1989) redescription of Caenoplana coerulea, a seminal vesicle is present in the genus.

Region 7 looks too wide to be part of the ejaculatory duct and, considering that the species lacks a permanent penis papilla, it seems that this region would probably constitute the external wall of the eversible penis. If that is the case, then it would be part of a male atrium and not of the ejaculatory duct. However, since the epithelium of the 7 regions was not described in the text, one cannot be sure about how to classify them. My suggestion is to include a description of the epithelium of those 7 regions and, if the epithelium of the final region (region 7) is more similar (or identical) to that of the common atrium than to the epithelium of more proximal regions of the duct, to reclassify it as a narrow male atrium and not as part of the ejaculatory duct. (see Winsor (1998) Aspects of taxonomy and functional histology in terrestrial flatworms (Tricladida: Terricola). Pedobiologia 42:412–432.)

I have only a few other small suggestions or questions to improve the text:

Page 2, line 56 (methods): change “contain” into “contained”
lines 60 and 61: replace “alcohol” with “ethanol”
Page 3, lines 114-116: common female duct has the abbreviation “cf” but the figures use “cd”, so either the text or the figures must be changed.
Page 4, line 152: I am not sure that the term “co-type” is still used. Wouldn’t that specimen be a paratype? Or, if it has the same status as the holotype, both would be syntypes.

---

## Round 0.2 · accepted · Accept

I see that you did not accept a few of the modifications suggested by the reviewers, but the paper seems to provide a good description of a new species, with anatomy, external morphology, and molecules, and that is enough to deserve immediate publication. This paper will be useful for future studies in Europe and elsewhere.